# Exploring the gut DNA virome in fecal immunochemical test stool samples reveals associations with lifestyle in a large population-based study

Paula Istvan [1,12], Einar Birkeland[1,12], Ekaterina Avershina [2,3], Ane S. Kværner[4], Vahid Bemanian[5], Barbara Pardini [6,7], Sonia Tarallo [6,7], Willem M. de Vos [8,9], Torbjørn Rognes [1,10], Paula Berstad[4] & Trine B. Rounge [2,3,11] ✉

Stool samples for fecal immunochemical tests (FIT) are collected in large numbers worldwide as part of colorectal cancer screening programs. Employing FIT samples from 1034 CRCbiome participants, recruited from a Norwegian colorectal cancer screening study, we identify, annotate and characterize more than 18000 DNA viruses, using shotgun metagenome sequencing. Only six percent of them are assigned to a known taxonomic family, with *Microviridae* being the most prevalent viral family. Linking individual profiles to comprehensive lifestyle and demographic data shows 17/25 of the variables to be associated with the gut virome. Physical activity, smoking, and dietary fiber consumption exhibit strong and consistent associations with both diversity and relative abundance of individual viruses, as well as with enrichment for auxiliary metabolic genes. We demonstrate the suitability of FIT samples for virome analysis, opening an opportunity for large-scale studies of this enigmatic part of the gut microbiome. The diverse viral populations and their connections to the individual lifestyle uncovered herein paves the way for further exploration of the role of the gut virome in health and disease.

Gut residing viruses represent an important component of the intestinal microbial ecosystem and may be collectively referred to as the gut virome. Recent large-scale efforts have shown the virome to comprise a vast and diverse population[1–5], of which bacteriophages (phages), i.e. viruses that infect and replicate in bacteria and archaea, make up the overwhelming majority. However, the extent of virome diversity in the gut remains poorly annotated, with only a minor fraction typically assigned taxonomy[2].

Viruses residing in the human gut are thought to act as a key modulator of the gut microbiome through their interaction with bacteria and the host immune system[6]. They may influence the structure and function of the bacterial community through facilitation of horizontal gene transfer[7], nutrient recycling, regulation of bacterial

[1]Centre for Bioinformatics, Department of Informatics, University of Oslo, Oslo, Norway. [2]Department of Tumor Biology, Institute of Cancer Research, Oslo University Hospital, Oslo, Norway. [3]Centre for Bioinformatics, Department of Pharmacy, University of Oslo, Oslo, Norway. [4]Section for Colorectal Cancer Screening, Cancer Registry of Norway, Norwegian Institute of Public Health, Oslo, Norway. [5]Pathology Department, Akershus University Hospital, Lørenskog, Norway. [6]Candiolo Cancer Institute, FPO-IRCCS, Turin, Italy. [7]Italian Institute for Genomic Medicine (IIGM), c/o IRCCS Candiolo, Turin, Italy. [8]Human Microbiome Research Program, Faculty of Medicine, University of Helsinki, Helsinki, Finland. [9]Laboratory of Microbiology, Wageningen University, Wageningen, The Netherlands. [10]Department of Microbiology, Oslo University Hospital, Oslo, Norway. [11]Department of Research, Cancer Registry of Norway, Norwegian Institute of Public Health, Oslo, Norway. [12]These authors contributed equally: Paula Istvan, Einar Birkeland. ✉e-mail: trinro@uio.no

virulence[8], and gain of antibacterial resistance[9]. Furthermore, viruses play a direct and indirect role in interactions between the human host and the bacterial community[10], and have been shown to exhibit temporal stability as high as that of their bacterial hosts[11,12].

The gut virome has been linked to human host and environmental factors, for specific food items[3,13] or viral populations[14], and like the bacterial community, its composition has been found to develop as a function of age[2]. The gut virome has also been associated with major chronic diseases such as inflammatory bowel disease and type 2 diabetes[15,16]. Dysregulation of gut bacteria and abundance of certain bacteria[17–19] are also proposed features of the association between the gut microbiome and colorectal cancer development[20]. These changes in the bacteriome are likely to be accompanied by phage dysregulation[21].

Given the high diversity and interindividual variability of the gut virome, large population-scale analyses are needed to decipher its role in human health and disease. Colorectal cancer screening programs, inviting millions each year, are currently running or in the planning stages in many countries across the globe[22]. A widely used screening strategy is based on fecal occult blood testing of gut samples, the fecal immunochemical test (FIT). The FIT is non-invasive, inexpensive, and scalable to large populations[23]. There is accumulating evidence that these gut samples are suitable for analysis of various features of the gut microbiome[24–26]. Combining the large numbers of gut samples from population-based screening programs with affordable shotgun metagenomics could propel unbiased and population-based virome studies.

To the best of our knowledge, no studies have yet been conducted analyzing the gut virome using FIT samples. With the availability of a large number of FIT samples collected in a Norwegian colorectal cancer screening trial, we have performed comprehensive profiling of the gut DNA virome. Here, we demonstrate suitability of FIT for virome studies. In addition, we describe viral diversity including taxonomy, genome integration, and functional potential, and assess associations of these factors with individual diet, lifestyle, and demographic factors.

## Results
### Dataset description
The study comprised 1640 individuals aged 55–76 who tested positive for FIT and were referred for colonoscopy within the Bowel Cancer Screening in Norway (BCSN) trial (Fig. 1a). DNA extracted from the samples was sequenced using shotgun metagenome sequencing and assembled into contigs, from which viral genomes were identified, dereplicated, and annotated (Fig. 1b). For details on the cohort description and data analysis, see Methods.

Raw shotgun metagenomic sequencing data comprised 13.5 billion paired-end reads, with 11.5 billion passing QC (median of 10.7 million reads per sample, IQR = 3.5 million; Fig. 2a). Storage time of samples before DNA extraction ranged from 34 to 1301 days, with a median of 198 days (Fig. 2a). Storage time did not impact DNA concentration, sequencing depth, assembly quality or the number of retrieved viral genomes ($|rho| \leq 0.05$, Fig. 2b). Spearman's rank correlation of DNA concentration to the sequencing depth, number of retrieved viral genomes, and alpha diversity ranged between $rho = 0.15$ and $rho = 0.18$, whereas correlation to the assembly quality was negligible ($rho = 0.04$, Fig. 2b). In total, we identified 1.7 million putative viral genomes, of which 3677 were classified as complete, 15,481 were classified as high-, and 30,484 were classified as medium quality, and were used in subsequent analyses (Supplementary Fig. 1). Overall, 18,268 of the 49,642 genomes (36.8%) were identified within host sequences, indicating a state of lysogeny. Clustering of viral genomes on a 95% similarity level resulted in 18,494 vOTUs (of which 1475 were comprised of genomes from 5 individuals or more; Supplementary Data 1), representing 37.3% of the potential vOTU diversity by Chao1

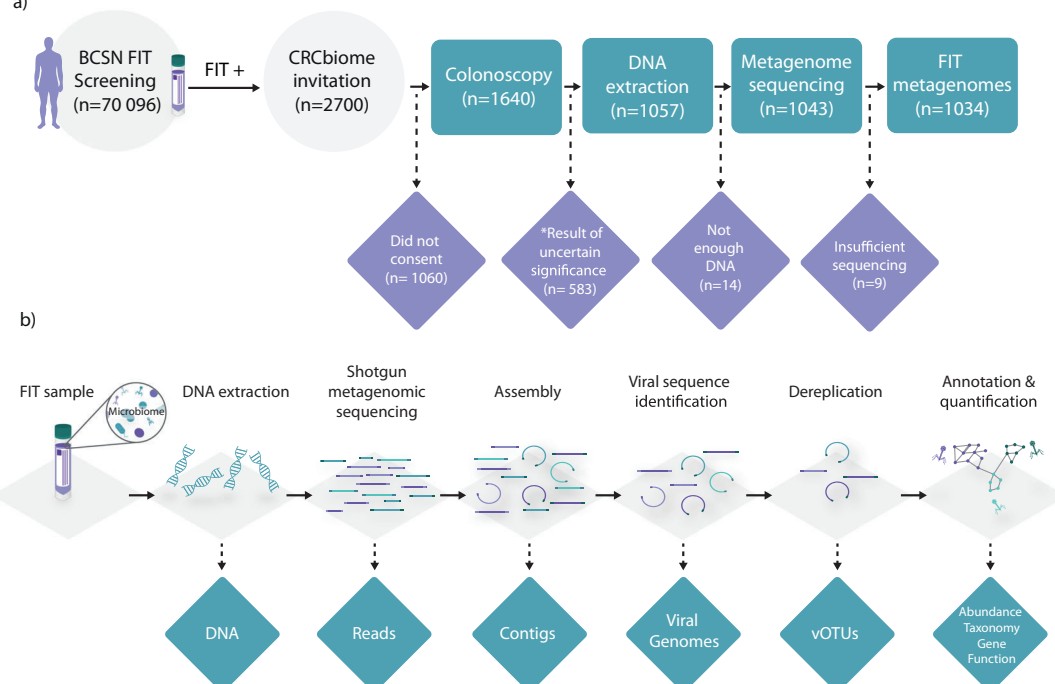

**Fig. 1 | Study design. a** participant flowchart. 2700 FIT-positive Bowel Cancer Screening in Norway (BCSN) participants were invited to the study. Excluded samples are indicated in purple. *Participants were excluded if they had findings of uncertain clinical significance, i.e., a low number of non-advanced adenomas or non-advanced sessile serrated lesions. **b** Workflow for virome characterization. DNA was extracted from the FIT leftover buffer. Shotgun metagenomic sequencing was performed on the Illumina platform and the resulting reads were assembled using metaSPAdes. Viral genomes were identified using Virsorter2, and then dereplicated using Galah. Representative vOTUs were taxonomically annotated using vConTACT2. DRAM-v was used for annotation of gene function. For details, see Methods. Created using Adobe Illustrator.

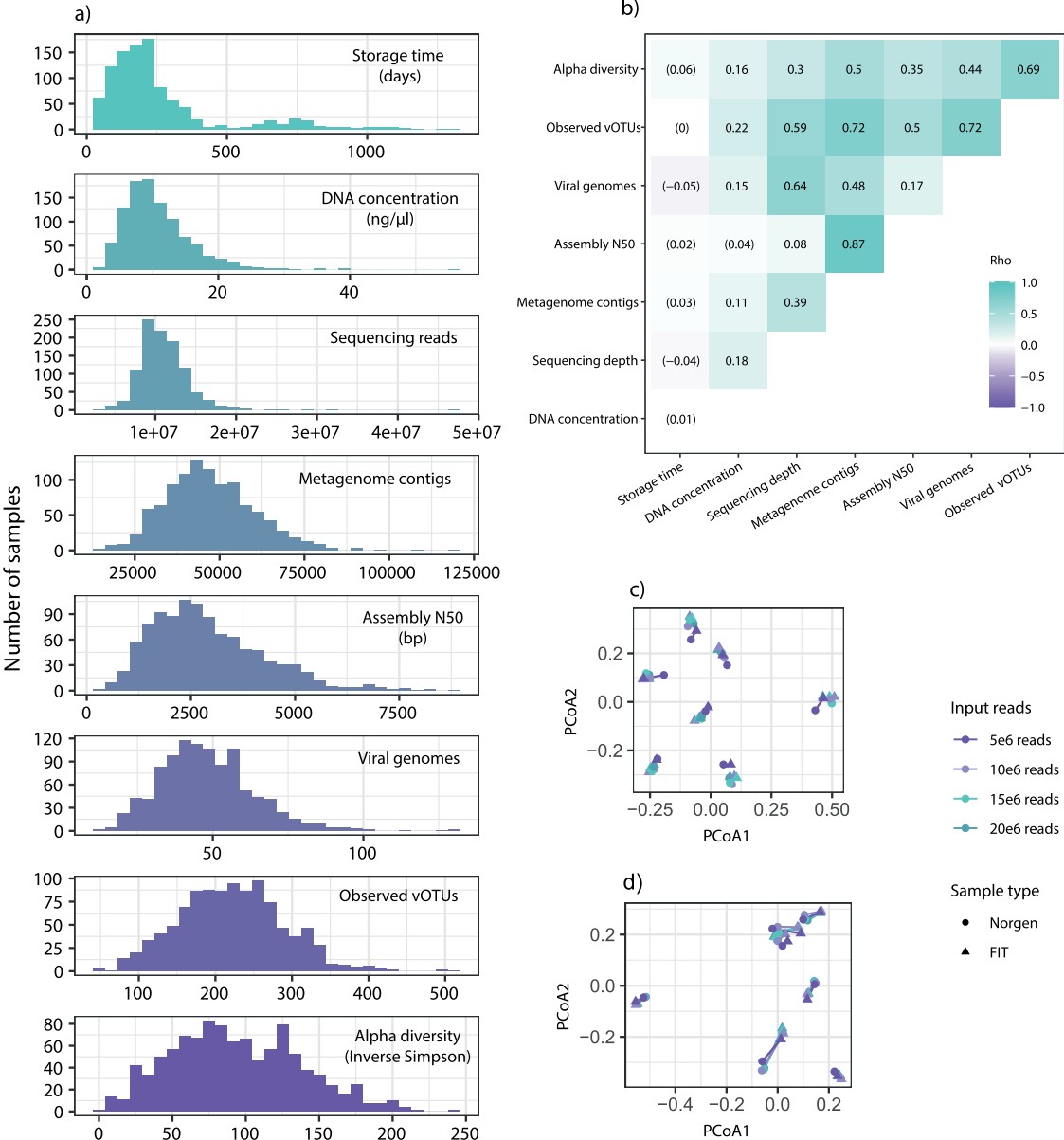

**Fig. 2 | Quality assessment of the virome dataset. a** Histograms of measures by sample including storage time, DNA concentration, number of sequencing reads, number of metagenome contigs, assembly N50, number of viral genomes, vOTUs observed after read mapping, and alpha diversity (inverse Simpson index). **b** Pairwise Spearman's rank correlation coefficients (rho) of the measures in (**a**). All correlations were statistically significant (FDR < 0.05), except for those with coefficients enclosed in parentheses. **c, d** Principal coordinate analysis (PCoA) of

**c** Jaccard distances derived from pairwise comparison of the identified viral genomes, and **d** Bray-Curtis distances derived from the abundance of CRCbiome vOTUs, in paired FIT and Norgen samples. Genomes with more than 95% ANI were considered to represent the same genome. Paired samples at the same level of subsampling are indicated by a connecting line, with the color representing the number of raw sequencing reads used as input, and with triangles representing FIT samples and points representing Norgen samples.

estimation of species richness. A mean of 223 vOTUs (sd = 69.3) per sample was observed after mapping sequencing reads to vOTU representative sequences (Fig. 2a). Inverse Simpson's diversity index ranged between 2.79 and 245 (mean = 93.5, sd = 43.7). With regards to beta diversity, the Bray-Curtis dissimilarity index ranged between 0.43 and 1 (mean = 0.84, sd = 0.065; Supplementary Table 1).

To assess the representativeness of FIT samples for the analysis of the gut virome, we performed a comparative analysis of seven paired fecal samples from an independent population, collected and stored using both FIT and Norgen nucleic acid kits, specialized for microbiome analysis. Both when assessing the identification of viral genomes and mapping reads from these samples to the CRCbiome vOTUs, we found sample identity to be more important than sampling methodology in determining the similarity of samples (PERMANOVA

$p_{sample\_id} = 0.001$, and $p_{sample\_type} =$ n.s. for both comparisons; Figs. 2c, d, respectively). Moreover, there were no significant differences in the number of viral genomes identified in FIT and Norgen samples (paired $t$-test, $p > 0.05$), nor between the paired samples and the CRCbiome FIT samples (Supplementary Fig. 2a). There was also no difference in the number of CRCbiome vOTUs detected between paired FIT and Norgen samples (paired $t$-test, $p > 0.05$; Supplementary Fig. 2b), nor any differences in the quality of genomes detected in FIT and Norgen samples (Supplementary Fig. 2c). Still, the paired samples displayed a lower number of observed CRCbiome vOTUs than the CRCbiome FIT samples, indicating that a significant fraction of the viruses detected in the CRCbiome cohort are specific to this population. By mapping sequencing reads from Thomas et al.[27] to the CRCbiome vOTUs, we found that the prevalence of CRCbiome vOTUs was somewhat lower in

this Italian population, but still corresponded well with those in the current cohort ($R^2 = 0.81$, $p < 0.001$; Supplementary Fig. 3).

## vOTU taxonomy and functional potential

Of 18,494 vOTUs, 6036 (32.6%) were assigned taxonomy based on their protein similarity to reference genomes in the phage-specific INPHARED database. An additional six vOTUs (0.03%) were clustered with eukaryotic viruses deposited in the Virus-Host database (Supplementary Table 2), with one being identified as human papillomavirus 6 (HPV6). This assignment was corroborated by mapping of reads from all subjects to the Papillomavirus Episteme database (PaVE)[28], indicating HPV6 to be present in one participant. Two conflicting reference genome assignments were found when comparing assignments made using the INPHARED database and the Virus-Host database. One vOTU clustered with the same reference genome in both the phage specific and the general virus database, with the latter indicating the virus to be infecting eukaryotes. However, by manual inspection, we found the host listed by the Virus-Host database to be erroneous, with the reference host reported in the original publication being bacterial[29] (Supplementary Table 2). A second vOTU was clustered with both a phage and a eukaryotic virus (*Acenitobacter* phage and an ameba virus targeting *Vermamoeba veriformis*, respectively), but while read mapping did not confirm the presence of either reference genome on a nucleotide level. Given limited viral databases, inconsistencies with host assignments, and the generally low prevalence of eukaryotic viruses in the gut[30], we further described taxonomy classification using the phage-specific INPHARED database only.

A majority of the phage vOTUs ($n = 4091$, 22.1% of all) were only assigned to a taxonomic order or class, and were more widely dispersed than family-annotated genomes (Fig. 3). The vOTUs that were assigned taxonomic family (1135), represented only 6.1% of all vOTUs. Overall, 19 viral families were identified. The most frequent viral family was *Microviridae* (Fig. 4a), with 528 members. Four families, and 416

vOTUs, of the order *Crassvirales* (*Suoliviridae*, *Intestiviridae*, *Crevaviridae*, and *Steigviridae*) were identified. In addition, the families *Peduoviridae*, *Inoviridae*, and *Winoviridae* were each identified with at least 20 members (Supplementary Table 3). A large fraction of genomes belonging to the class *Caudoviridicetes* belonged to lineages with the former morphology-based classifications *Siphoviridae*, *Myoviridae*, and *Podoviridae* ($n = 2849$). The fraction of uncovered vOTU diversity, according to Chao1 estimates, differed by family, with 60% and 74% of *Crevaviridae* and *Winoviridae* respectively, being detected. On the other hand, the detection rates of *Microviridae* and *Inoviridae* were much lower, with 9.9% and 7.3% identified respectively (Supplementary Table 3). Multiple vOTU characteristics differed markedly between viral families, including genome size (Fig. 4b), genome integration (Fig. 4c), gene annotation frequency (Fig. 4d), and the rate at which auxiliary metabolic genes (AMGs) were detected (Fig. 4e).

*Intestiniviridae*, *Suoliviridae*, *Steigviridae*, and *Inoviridae* genomes were almost exclusively identified as unintegrated (Fig. 4c; Supplementary Table 4), while genomes of the *Crevaviridae* and *Microviridae* families had a small, but not insignificant, fraction of integrated genomes. On the other hand, most genomes of the *Peduoviridae* and *Winoviridae* families were identified in an integrated state.

AMGs were detected in 24.3% of vOTUs, being more commonly detected in *Crassvirales* (67.5%), and less common in *Microviridae* vOTUs (1.1%). AMGs from Organic nitrogen and Miscellaneous (MISC) functional groups were detected in 12.8% and 11.7% of vOTUs, respectively, being about five times more prevalent than any other functional group or combinations of these (Supplementary Fig. 4). On a family level, the prevalence of the Organic nitrogen group of AMGs was almost absent from vOTUs belonging to *Crassvirales* (0.2%), being largely confined to genomes classified as belonging to the *Peduoviridae* family and to genomes without a family annotation (Fig. 4e). AMGs of the MISC group (almost exclusively genes related to pyrimidine deoxyribonucleotide synthesis) were detected in a majority

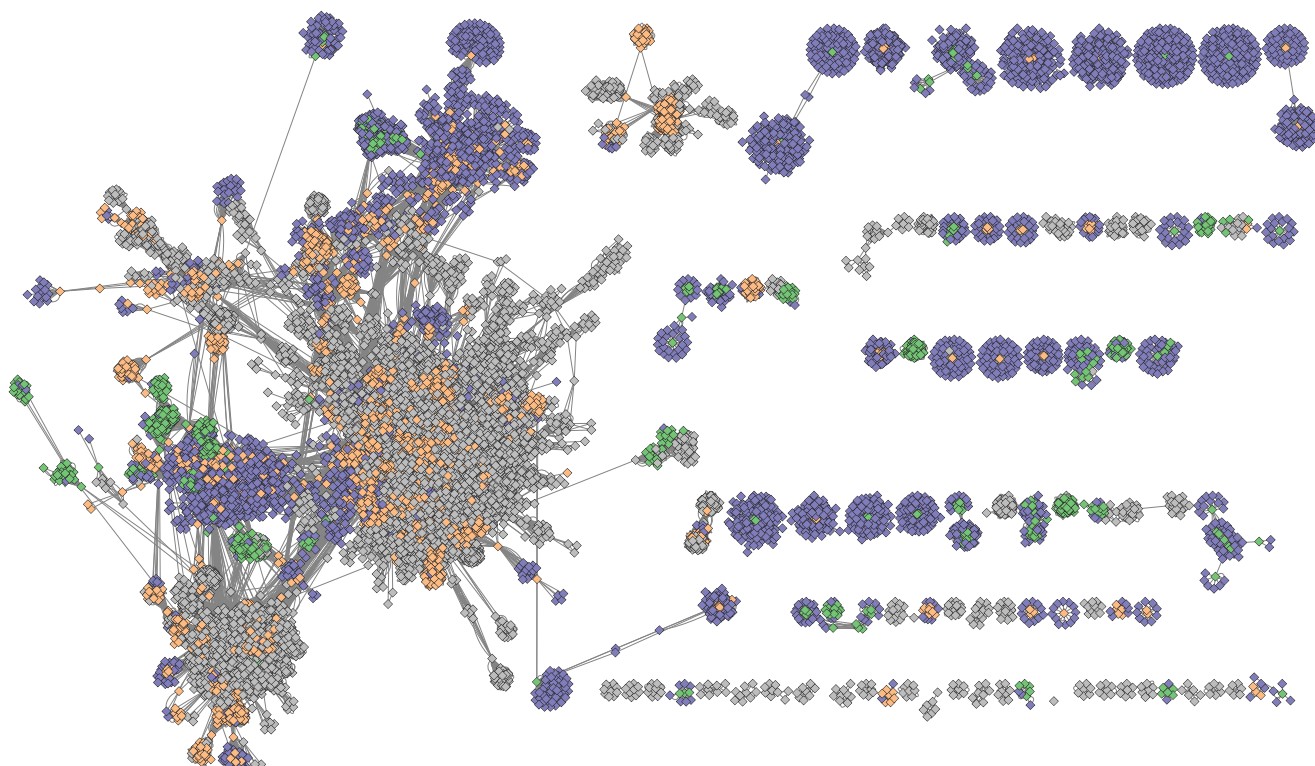

**Fig. 3 | Clustering of the vOTUs based on their gene similarity on a protein level.** Green - vOTUs that had taxonomic family annotation; orange - vOTUs that were assigned taxonomic order, but not family; gray - vOTUs with no taxonomic assignment; purple - reference viral genomes. Outlier vOTUs (those with no significant associations) were excluded from visualization.

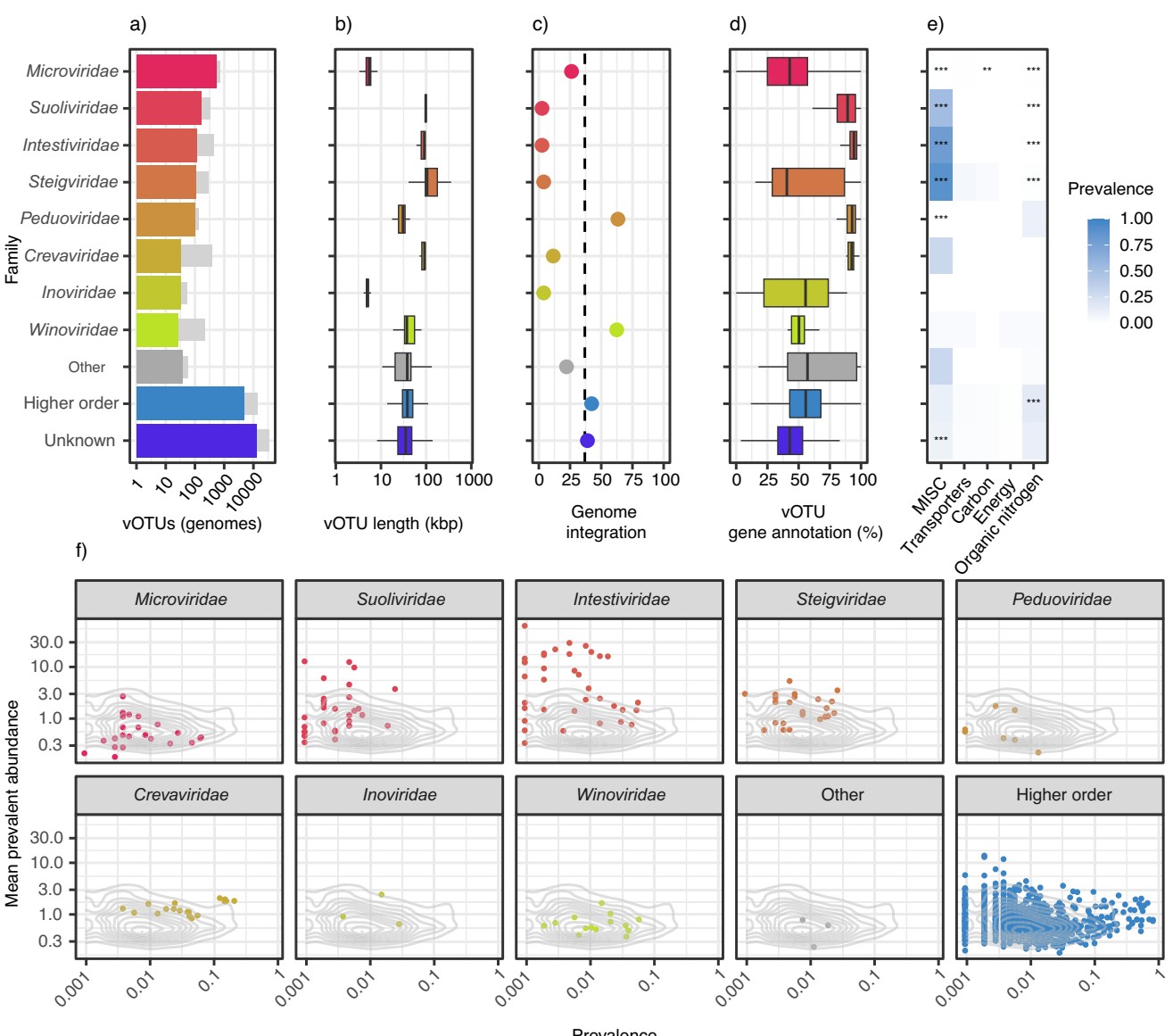

**Fig. 4 | Genome annotation and population distribution. a** Taxonomic classification of vOTUs at the family level. The vOTUs belonging to families with fewer than 20 representatives are categorized as "other". The "unknown" group constitutes those not clustering with any reference genomes, whereas those clustering with reference genomes annotated at higher levels are labeled "higher order". Light gray bars indicate the total number of genomes (pre-dereplication) according to the taxonomic assignment of their representative vOTUs. **b** Genome size distribution for genomes belonging to each taxonomic category. For stratification by completeness, see Supplementary Fig. 1. **c** The percentage of viral genomes classified as integrated. The dashed line represents the overall percentage of integrated genomes. See Supplementary Table 5 for details. **d** Percentage of annotated genes per vOTUs according to viral family. **e** The fraction of genomes carrying genes annotated with AMGs by AMG category and family. Asterisks indicate significant deviations in AMG category prevalence for one family when compared to the rest (post-hoc two-sided Fisher exact test, *$p < 0.05$, **$p < 0.01$, ***$p < 0.001$; p-adjustment by Bonferroni). MISC Miscellaneous, Carbon Carbon utilization. **f** Prevalence and mean abundance (if detected) for vOTUs with at least 2 constituent genomes by taxonomic assignment. The 2D density contour lines indicate the overall distribution of prevalence and abundance for vOTUs (≥2 constituent genomes). In **b** and **d** the borders of the boxes span the first (Q1) to third (Q3) quartiles, with the middle line representing the median. Whiskers extend to the most extreme point in the dataset but not further than Q1-1.5IQR (lower limit) and Q3 + 1.5IQR (higher limit). Outliers are shown as individual points.

(67.1%) of the *Crassvirales* vOTUs, and in particular those belonging to *Steigviridae* (78.8%) and *Intestiviridae* (88.1%).

Abundance was assessed by mapping reads from all samples to each vOTU. This increased the total number of detected viruses in each sample (mean identified genomes per sample 48; mean observed vOTUs 215). Out of 18,494 vOTUs, 2576 were detected in ≥1% of the population. A mean of 24.4% of viral abundance by sample were attributed to vOTUs with any taxonomic annotation (range 7.9–83.0%; Fig. 4f). *Crassvirales* vOTUs were detected in 70.6% of samples and constituted up to 75.4% of viral abundance (median 0.6%). Overall,

*Crassvirales* vOTUs, and especially those of the *Intestiviridae* family, were more abundant when detected, whereas *Microviridae* and *Peduoviridae* were less abundant.

**The gut virome reflects individual health-related lifestyle, including smoking, physical activity, and carbohydrate intake**
We assessed differences in virome alpha and beta diversity to determine how the gut virome varied by diet, lifestyle, and demography. Out of 25 selected variables (Supplementary Table 5), we identified 9 significant associations with alpha diversity as measured by the

inverse Simpson's index (Fig. 5a; Supplementary Data 2). Among these, the largest effect sizes were found for physical activity (positive association), alcohol consumption (positive association), and dietary carbohydrate consumption (negative association). Viral beta diversity was significantly associated with 17/25 variables assessed (Fig. 5b; Supplementary Table 6), with several being health-related lifestyle factors. Indeed, the strongest association was observed for a composite HLI, with other lifestyle variables being relatively strongly associated, including dietary fiber consumption, physical activity, and smoking, among others. Assessing the differential abundance of individual vOTUs, we identified several representative genomes being associated with the same set of variables (Fig. 5c; Supplementary Data 3). Here, the highest numbers of differentially abundant vOTUs were found for smoking and physical activity (Fig. 5d). Dietary fiber consumption was also associated with a high number of differentially abundant vOTUs (Fig. 5d, Supplementary Fig. 5). Among differentially abundant vOTUs, there was no skew in the frequency of any viral families, nor with the frequency of viruses with a lytic or lysogenic lifestyle. On the other hand, we observed a clear over-representation of AMGs across the differentially abundant vOTUs (Supplementary Fig. 6), especially for those related to smoking. Due to the inclusion of participants from a high-risk screening population, there was an over-representation of colorectal cancer. To assess whether this might have influenced the observed associations, we performed sensitivity analyses excluding any participants with colorectal cancer and found no overall differences in identified associations (Supplementary Fig. 7).

Overall, 69 vOTUs were related to at least one lifestyle or demographic variable, with 22 being associated with multiple. As an example, one vOTU (CRCbiome_vOTU05693, no taxonomic assignment) was negatively associated with smoking, and positively correlated with physical activity and dietary fiber consumption (Fig. 5d). This vOTU was identified in 62.2% of participants, and was representative of 23 viral genomes, none of which were found to be integrated in a host genome. Gene annotation (44% of predicted genes) identified genes encoding an integrase, a DNA topoisomerase, and two methyltransferases (Fig. 5e), indicating a potential capacity of this vOTU to integrate a bacterial host genome. DNA methylase, which is crucial for host defense and epigenetic regulation, was also identified in the CRCbiome_vOTU05693 genome.

## Discussion

The gut microbiome, and the gut virome in particular, has largely been studied using either fresh stool samples or stool samples preserved in buffers designed for snap-shot stabilization of the microbiome[31]. Here we show that the analysis of the gut virome using samples collected in a routine setting and stored in a FIT buffer designed for hemoglobin stabilization is feasible. The reliability of the FIT sampling kits in the analysis of bacteria has repeatedly been demonstrated[24,32,33], but to the

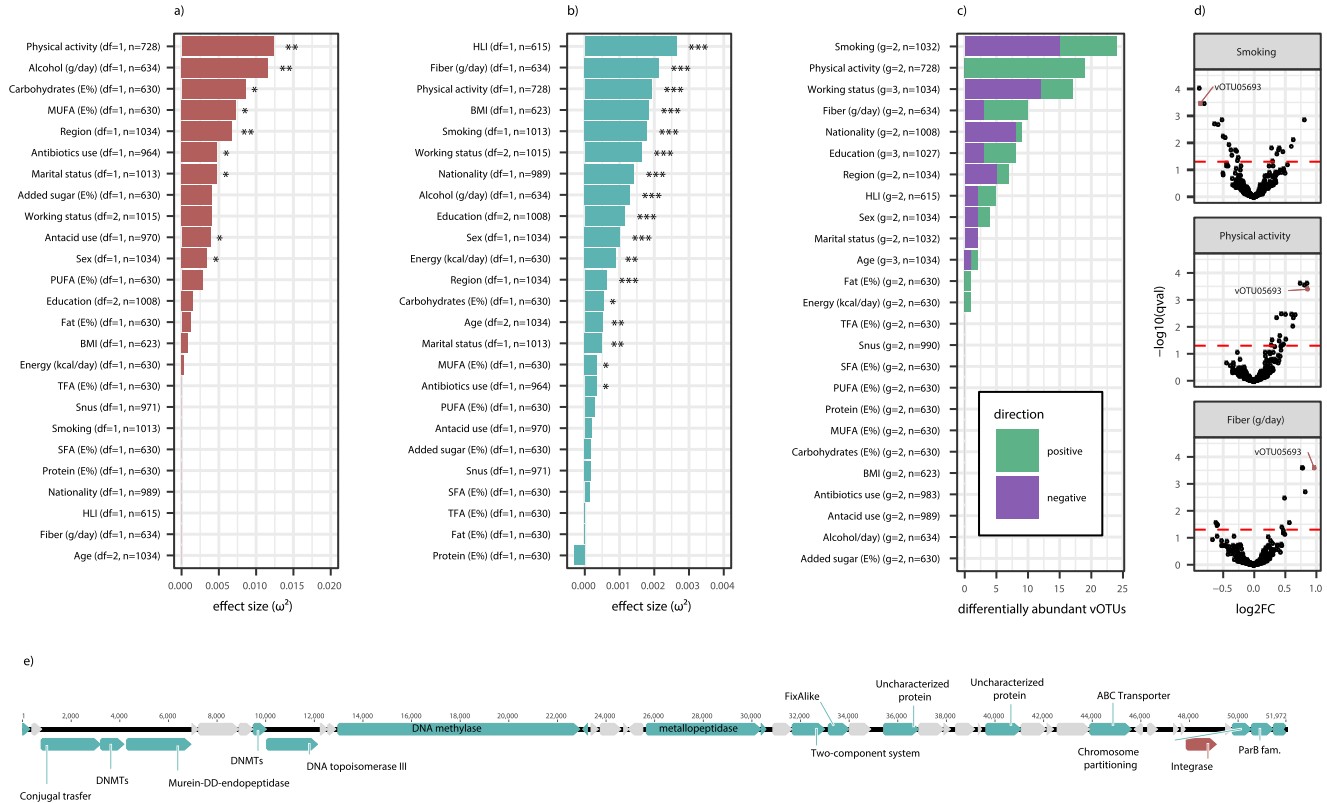

**Fig. 5 | Associations of viral diversity with diet, lifestyle, and demographic variables. a** Effect sizes of alpha diversity of vOTU abundance as measured by the inverse Simpson index by two-sided ANOVA. **b** Effect sizes of associations between vOTU beta diversity (Bray-Curtis index) by PERMANOVA. Effect sizes for alpha and beta diversity are derived using the omega-squared measure from ANOVA tests of the association between diversity measures and each variable, with correction for sample sequencing coverage. *$p < 0.05$, **$p < 0.01$, ***$p < 0.001$; exact $p$-values are given in Supplementary Data 2 and Supplementary table 6. **c** Number of significantly differentially abundant vOTUs identified by MaAsLin2, colored by direction of association. For continuous variables, the top and bottom tertiles were

compared. Details are available in Supplementary Data 3. **d** Volcano plots showing the relationship between effect size (log2 fold change) and significance level ($q$-value) for vOTUs for physical activity, smoking, and fiber intake, from top to bottom. The red dotted line indicates the significance threshold. MUFA monounsaturated fatty acids, PUFA poly-unsaturated fatty acids, TFA trans fatty acids, SFA short-chain fatty acids, BMI body mass index, HLI healthy lifestyle index. **e** Genomic map representation of CRCbiome_vOTU05693, associated with smoking, physical activity, and dietary fiber intake, with predicted genes with annotations in green, without annotations in gray, and integrase gene annotation highlighted in red.

best of our knowledge, the present study is the first to demonstrate this for viruses. The use of FIT samples enabled an in-depth characterization of the viral constituents of the human gut and allowed us to discern associations between the gut virome and important health-related lifestyle factors, although interpretation of findings remains hampered by the incompleteness of reference databases.

Our analysis of paired FIT and Norgen samples demonstrated that the use of FIT kits does not entail a significant loss of viral diversity. Even though FIT samples are designed to capture as little as 10 mg of fecal matter, only a minor fraction of samples (<1%) failed to produce sequencing data, and viruses were identified in all samples with sufficient data. Stability under storage conditions and DNA quality and quantity are key for the reliability of generated data. Our finding that DNA concentration, sequencing depth, and viral diversity were only negligibly affected by sample storage duration lend support to the use of FIT kits as a suitable sampling methodology for virome characterization. FIT sampling is widely employed in population-based colorectal cancer screening programs, highlighting the potential for large-scale virome studies across the world.

In this extensive analysis of the gut virome in 1034 Norwegian adults, we identified over 18,000 vOTUs representing more than 49,000 complete, high- or medium-quality viral genomes detected across the population. Despite a large sample size for a relatively homogeneous population, our estimates of species richness show that increased sampling would be required to more fully describe the gut virome in this setting. Moreover, due to the exclusive measure of DNA as a source of genetic information, our analyses do not include RNA viruses. Still, the uncovered viral diversity is substantial, and is in line with studies using microbiome-adapted sampling methodology[2,3]. Similar to other reports[2,3,5,12], two-thirds of the vOTUs detected in our study were not represented in current state-of-the-art reference databases, with only four vOTUs being assigned to eukaryotic viruses, one of which was human papillomavirus 6. Furthermore, only one-fifth of those bacteriophages that were represented, were assigned taxonomy at the level of family, clearly demonstrating the lack of data on the human virome. Using the recently ratified taxonomy[34], we found *Microviridae* to be the most commonly assigned viral family among the vOTUs, with most *Microviridae* vOTUs being representative of a small number of genomes. On the other hand, vOTUs annotated as *Crevaviridae*, one of the families belonging to *Crassvirales* order, consisted of significantly larger clusters of genomes, indicating that a larger fraction of *Crevaviridae* genomes were identified when compared to *Microviridae*. This finding of a highly diverse group of *Microviridae* vOTUs is in line with current understanding of this viral family; the high rate of mutations and recombination in their characteristically small genomes not only facilitates rapid evolution and adaptation, but also leads to high intra-family diversity[35].

Along with *Crevaviridae* viruses, other viruses of the *Crassvirales* order displayed lower diversity, and, except for the *Steigviridae* viruses, had a higher fraction of genes annotated. Viruses of the *Steigviridae* family have likely followed an independent evolutionary path from other *Crassvirales* viruses, potentially acquiring novel genes and functions via mechanisms like horizontal gene transfer[36]. Other observed characteristics of the *Crassvirales* viruses such as their size (97–131 kb), almost exclusively lysogenic nature, and high prevalence and abundance, are consistent with other studies[14,37].

We found about a third of viral genomes to be integrated with the genome of its host. Genome integration is a common manifestation of lysogeny, employed by temperate viruses. Lysogeny is one of two predominant viral lifecycles, with the other being the lytic one[38]. The lytic cycle involves viral replication, resulting in host cell destruction and the release of new viruses. In contrast, the lysogenic cycle represents a dormant state, wherein the viral genome is replicated in sync with its host, often being integrated into the host genome, creating a prophage which can be activated to revert to the lytic cycle under certain conditions. Strategies for the study of phage lifecycles include the identification of phages with a potential for transition to a lysogenic state, and direct detection of host genome insertion[39,40]. The former of these is hampered by poor database coverage, and does not provide a measure of actual lysogeny, whereas the latter, which we employed, does provide such a measure, but does not count phages whose lysogenic state occurs in a rolling cycle replicating or plasmid-like state within the host cell. There were clear differences between viral families in their propensity for genome integration, where in contrast to the almost exclusively lytic *Crassvirales* and *Inoviridae* viruses, two viral families, *Peduoviridae* and *Winoviridae*, contained mainly prophages. Interestingly, in a recent study on prophages in infants and adults, *Peduoviridae* was among the most frequently detected, whereas *Winoviridae* phages were not listed[41].

Auxiliary metabolic genes (AMGs) are important for phage modulation of bacterial function[42]. The two most common AMG categories identified in the current population included nitrogen metabolism and nucleotide synthesis (pyrimidine deoxyribonucleotide synthesis, or MISC in Fig. 4e). These AMGs can enhance viral replication efficiency by boosting the bacterial host's pyrimidine synthesis, providing a selective advantage to the virus. This could disrupt the bacterial host's pyrimidine balance, leading to potential cell resource misallocation, nucleotide overproduction, or DNA damage. The small genomes of the *Microviridae* contained few AMGs. In general, when detected, viral genomes tended to contain multiple AMGs per genome. AMGs were common in *Crassvirales* vOTUs, with nucleotide synthesis genes being over-represented and organic nitrogen AMGs being under-represented. Genes involved in metabolism of organic nitrogen were primarily found in the *Peduoviridae* family and within vOTUs that remained unclassified at the family level.

Lifestyle factors have been shown to exhibit significant associations with the bacteria of the gut[43]. However, far less is known for the viral fraction. We conducted a comprehensive analysis of how viral abundance was related to individual diet, lifestyle, and demographic factors, measured in broad and generalizable terms. Virome alpha diversity displayed some variation, but not as pronounced as the beta diversity. We found lifestyle factors such as physical activity, dietary fiber, and alcohol consumption to have consistent associations with gut virome alpha and beta diversities. Although differences in lifestyle assessment and categorization make direct comparisons difficult, recent studies of various populations have found alcohol intake, as well as diets reflecting a higher intake of fiber to be associated with virome characteristics[3,13,14], while no associations were found for physical activity. Smoking has been extensively studied for its genetic and epigenetic effects in human cells[44,45]. We found smoking to be associated with beta diversity, in line with some[3], but not all[13] prior reports. Contrary to what has been reported previously[2], we did not find an association between gut virome composition and participant age. While the generalizability of our results could be restricted by the age selection of the study population, the results are in line with a recent report showing maintained diversity in subjects of advanced age[46].

Consistent with beta diversity differences, individual vOTUs were differentially abundant according to subject lifestyle. Differentially abundant vOTUs displayed no propensity towards particular viral clades, nor genome integration state, but we did observe an intriguing over-representation of AMGs, particularly for vOTUs associated with smoking. Notably, we found that several of them were differentially abundant with regard to a number of diet, lifestyle, and demographic factors. Moreover, an index capturing multiple aspects of a healthy lifestyle (healthy lifestyle index; HLI) was found to have the largest effect size in relation to gut virome beta diversity. This suggests that several lifestyle factors that affect health may act in concert to shift virome composition. There has been a recent trend in public health research focusing on the overall pattern of lifestyle choices, rather than individual factors[47].

An example illustrative of the challenges and promise of gut virome analyses was our identification of CRCbiome_vOTU05693 as being negatively associated with smoking, and positively associated with physical activity and dietary fiber intake. While being a possibly important indicator of a health-associated lifestyle, no taxonomic information was possible to derive from current reference databases. None of the annotated genes were AMGs, but indicated a capacity for host genome integration, host defense, epigenetic gene regulation, and maintenance of genome stability[48]. Still, none of its 23 constituent genomes were identified in an integrated state. These observations highlight the need for continued studies and expansion of reference databases for the gut virome, and functional studies of particular viruses.

Collectively, the associations indicate that lifestyle choices may influence the composition and viral make-up of the gut virome. While the evidence is limited, recent intervention studies have shown that a short-term change of diet can lead to significant alterations in both the human and mouse gut virome[11,49]. It is likely, though, that alterations in viral abundances are accompanied by, or even precipitated by shifts in abundance of their bacterial hosts.

The main strength of this study includes a large population, which draws on participant recruitment carried out as part of a population-based Norwegian screening trial, inviting all residents of a defined age range and geographic region[50]. Standardized data collection included rich and high-quality data on participant diet and lifestyle. Minimal technical interference in the high-quality metagenomes enabled detailed analyses of virome taxonomy, annotation, and lifecycle. Comprehensive analyses of alpha and beta diversity, vOTUs differential abundance, and the nuances between them, provide a multi-faceted depiction of the virome. Despite these strengths, there are limitations to consider. The participants had a FIT positive test, meaning that they had traces of blood in their stool samples. Therefore, the proportion of individuals with premalignant or malignant colorectal cancer lesions was higher than in the general population. Sensitivity analyses excluding participants with a malignancy did not, however, impact the study outcomes.

This study shows that the virome can be reliably profiled using FIT samples, by identifying more than 18,000 vOTUs from over 1000 individuals and identifies the virome as being deeply connected to host lifestyle and demography. The associations between the gut virome and subject lifestyle suggests a potential for the gut virome to serve as a source of biomarkers. While microbiome studies have identified gut bacteria as disease biomarkers[51], the development of viral biomarkers will require large-scale studies defining sources and measures of gut virome variation.

## Methods

### Study population
The CRCbiome project was approved by the Norwegian Regional Committees for Medical and Health Research Ethics (Approval no.: 63148). The MITOS cohort project was approved by the local Ethics committee (AOU Città della salute e della Scienza di Torino, Italy; Approval no.: 0061857). CRCbiome enrolled individuals aged 55–76 who tested positive for FIT (and were referred for colonoscopy) between October 2017 and March 2021 from the Bowel Cancer Screening in Norway (BCSN) trial, which is a population-wide randomized trial comparing the effectiveness of once-only sigmoidoscopy and biennial FIT testing. Out of the 2700 individuals invited to participate, 1640 met the inclusion criteria and provided informed consent. Participants were not compensated. Details on recruitment procedures can be found in Kværner et al.[50]. All participants provided FIT samples (Eiken Chemicals Ltd., Tokyo, Japan) containing fecal matter that were self-collected at home and shipped to the laboratory by mail at ambient temperature. Following FIT testing, samples were stored at −80 °C until withdrawal of leftover buffer from the FIT container

(~1600 μl; containing about 10 mg fecal matter) and DNA extraction (see details below). For the purpose of the CRCbiome overall aim, samples were selected based on their colonoscopy results, excluding those without colonoscopy, or with findings of uncertain clinical significance. The availability of sufficient DNA ( > 0.7 ng/μl) and metagenome data (>1 gigabase after QC) was also required. The final number of FIT metagenomes included in the study was 1034 (Fig. 1a) and participant characteristics are detailed in Supplementary Table 5.

To assess the representativeness of FIT sampling for virome analyses, we included FIT leftover samples paired with stool samples collected in nucleic acid collection and transport tubes with RNA stabilizing solution (Norgen Biotek Corp., ON, Canada), hereafter referred to as Norgen samples, from 7 Italian individuals. These individuals were recruited in the frame of the regular Piedmont Region CRC screening in the Microbiome and MiRNA in Torino Screening (MITOS) project[26,52]. Within the screening program, all residents, aged 59–69 are invited to undergo a single sample biennial FIT (Eiken). Stool for the Norgen samples was collected at home before the appointment for colonoscopy and before the bowel preparation. The Norgen samples were brought to the hospital the day after the collection and immediately frozen at −80 °C until DNA extraction. FIT samples were stored at −80 °C until use. For this work, samples were analyzed in an anonymized manner.

### Questionnaire data
Prior to the colonoscopy, participants were asked to complete two questionnaires on diet, lifestyle, and demography: a food frequency questionnaire (FFQ), developed and validated[53–55] at the Department of Nutrition, University of Oslo, and a lifestyle and demographic questionnaire (LDQ), developed in-house. The FFQ is designed to capture the habitual diet during the preceding year. The current questionnaire version includes a total of 23 questions, covering 256 food items. For each food item, participants were asked to record frequency of consumption, ranging from never/seldom to several times a day, and/or amount, typically as portion sizes given in various household units. Dietary intake was calculated using the food and nutrient calculation system, KBS, developed at the Department of Nutrition, University of Oslo, with its associated database, which is largely based on the Norwegian Food Composition Table[56]. We focused on key dietary measures, including total energy intake (kcal/day), intake of macronutrients (in g/day or energy percentage (E%)), and selected food groups (g/day), being linked to risk of major chronic diseases such as cancer (described in further detail below)[57,58]. The FFQ also included questions on body weight (kg) and height (m), which was used to calculate participants' BMI (kg/m$^2$). The LDQ is a questionnaire developed specifically for the CRCbiome study to obtain data on key lifestyle and demographic variables. The questionnaire includes ten questions in total, where the ones relevant to the current study included demographic factors (national background, education, occupation, and marital status), antibiotic and antacid usage during the last three months, smoking and snus habits, and physical activity level. In the question concerning tobacco usage, participants were asked about their current habits, including the daily number of cigarettes/snus portions, and to recall years since possible cessation and total years of use. In the present study, smokers and snusers were defined as self-reported regular or occasional users, or those being registered with recent use (<10 years). For physical activity, participants were asked to report the time spent in low, moderate, and vigorous physical activity per week during the past year. Total amount of moderate to vigorous physical activity (min/week) was calculated by summing the time spent in moderate and vigorous activity, the latter weighted by a factor of two to best match national[59] and international recommendations[60,61].

As a measure of the overall diet and lifestyle, we created a healthy lifestyle index (HLI), grading participants by adherence to the

following seven recommendations (primary intended to prevent cancer, but is also relevant for other major chronic disease): (1) be a healthy body weight, (2) be physically active, (3) consume a diet rich in whole grains, vegetables, fruit, and beans, (4) limit intake of "fast foods" and other processed foods high in fat, starches, or sugars, (5) limit consumption of red and processed meat, (6) limit consumption of sugar-sweetened drinks, and (7) limit alcohol consumption. Further details on the HLI can be found Kværner et al.[62] and Supplementary Table 5.

### Sample collection, library generation, and metagenome sequencing

Following collection of FIT sampling kits and measurement of fecal occult blood concentration, leftover buffer was used as input material for DNA extraction and library preparation for the generation of shotgun metagenome sequencing data. DNA was extracted using the QIAsymphony automated extraction system (Qiagen, Hilden, Germany) using an off-board lysis protocol described in Kværner et al.[50]. Extracted DNA was tagmented, indexed and amplified according to the Nextera DNA Flex Library Prep Reference Guide (Illumina, CA, USA), except scaling down the reaction volumes to one-quarter of the reference. Indexed DNA fragments from each sample were then combined into library pools, each comprising 240 samples, and size selected to a fragment size of 650–900 bp using AMPure XP (Beckman Coulter, IN, USA). Sequencing was performed on the Illumina NovaSeq system using S4 flow cells with lane divider, with each pool sequenced on a single lane resulting in paired-end $2 \times 151$ bp reads. Shotgun metagenome sequencing was performed aiming to achieve 3 gigabases per sample. A schematic of the wet-lab workflow is presented in Supplementary Fig. 8. As controls, we included six negative controls for DNA extraction and a further two negative controls for library preparation. The DNA extraction controls resulted in the generation of a total of 32 QC sequencing reads, whereas sequencing of the library preparation negative controls resulted in a total of 3 reads. Given the negligible number of reads in the negative controls, additional contaminant removal procedures were not considered.

For the stool in paired FIT and Norgen kits, DNA extraction was performed using a DNeasy PowerSoil Pro Kit (Qiagen, Hilden, Germany) according to the instructions of the manufacturer and starting from 200 μl of fecal samples. DNA was eluted in 50 μl of the elution buffer provided with the kit. The DNA quantification was performed with a Qubit DNA high-sensitivity assay kit (Thermo Fisher Scientific, MA, USA). Sequencing libraries were prepared using an Illumina DNA Prep kit (Illumina, CA, USA), following the manufacturer's guidelines and a protocol described in Thomas et al.[27]. Sequencing was performed on the NovaSeq6000 (Illumina, CA, USA) at the internal sequencing facility of the Italian Institute for Genomic Medicine. To facilitate comparison at relevant sequencing depths, raw sequencing reads from paired FIT and stool samples were randomly subsampled to 5, 10, 15, and 20 million paired-end reads per sample.

### Sequence reads quality control and assembly

The metagenome processing framework Metagenome-ATLAS[63] was used for sequencing quality control and assembly. In brief, ATLAS employs BBTools[64] utilities for adapter and quality trimming of reads, and for the removal of human genome and PhiX reads. Quality trimmed reads, both paired and unpaired, were used for de novo assembly using metaSPAdes[65]. For information on versions of all bioinformatics tools and databases used, see Supplementary Table 7.

### Viral sequence identification, dereplication, quantification, and assessment of genome integration

Viral genomes were classified with VirSorter2[66] using the default database and parameters, and with metagenomic contigs >1500 bp as input. CheckV[67] with the default database was used for the assessment of genome completeness, quality, level of host sequence content, annotation of host genome integration, and to extract the fractions of contigs determined to contain viral sequences. Viral genomes assigned a quality of medium or higher (corresponding to >50% completeness) by CheckV assessment were extracted and considered for further analysis. We clustered viral genomes by average nucleotide identity (ANI) to define viral operational taxonomic units (vOTUs), or clusters using the dereplication tool Galah[68], defining clusters by an ANI threshold of 97% covering at least 70% of each genome's length. The viral genome with the highest completeness in each cluster was chosen as the representative genome for that vOTU. Quality-controlled paired-end reads from all participants were mapped to each vOTU using BBMap[64], with the following options: *pairlen = 1000, pairedonly = t, minid = 0.9, maxindel = 100, ambiguous = all, maxsites = 10*. The vOTU coverage was calculated using the *pileup* function from BBTools, and vOTU abundance was recorded as the median coverage for those with reads mapping to at least 75% of the genome. Sequencing data from paired FIT and Norgen samples were subjected to the same procedure for assembly and viral genome identification, with an additional dereplication analysis including viruses identified at all different levels of subsampling. Quality-controlled reads were also mapped to the representative genomes identified in the CRCbiome cohort. Mapping of reads to CRCbiome vOTUs was also performed for sequencing data from a publicly available dataset[27].

### Annotation of viral genomes

Taxonomic classification of vOTUs was carried out using vConTACT2[34], based on proteins identified with Prodigal[69]. The reference database for phage genomes was established using INPHARED[70], which retrieves and filters GenBank phage genomes, constructing a database exclusively comprised of complete or near-complete genomes. We additionally used the Virus-Host DB database[71] that covers RefSeq and GenBank deposited viruses and includes manually curated information on host identity retrieved from GenBank, RefSeq, UniProt, ViralZone, and literature surveys, to identify eukaryotic viruses. vConTACT2 uses a network-based approach to identify viral clusters based on viral protein sequences. For processing of vConTACT2 clustering, graphanalyzer[72] was used. Here, taxonomy was assigned if a vOTU had a direct or indirect connection (up to one degree removed) to a reference, where the strength of the connection prioritized the taxonomy assignment. For phage assignment, vConTACT2 was run with parameters *--db 'ProkaryoticViralRefSeq94-Merged' --rel-mode 'Diamond' --pcs-mode MCL --vcs-mode ClusterONE*. When searching for eukaryotic viruses, the *--db* setting was set to *'None'*. Cytoscape[73] was used to visualize the vOTU network excluding vOTUs with no significant associations (outliers). Two replicates of a community standard (ZymoBIOMICS Microbial Community Standard) were sequenced and processed, resulting in identification of 15 proviruses annotated as phages specific to the bacteria included in the community standard (Supplementary Table 8).

DRAM-v[74] was employed for gene annotation of vOTUs, using the databases Pfam[75], VOGDB[76], KOfam[77], dbCAN[78], and RefSeq[79]. Auxiliary metabolic genes (AMGs) were defined using default settings in DRAM-v. The prevalence of AMGs was calculated by presence/absence of each category of AMG per vOTU.

### Statistics

To evaluate differences in the number of viral genomes identified or observed vOTUs after mapping to CRCbiome vOTUs in paired FIT and Norgen samples, we performed paired *t*-tests, employing data resulting from subsampling to 15 million reads per sample. Comparisons between paired samples and CRCbiome samples were carried out using a linear regression model, adjusting for sequencing depth. The R package *vegan*[80] was used to calculate alpha diversity (inverse Simpson

index), with between-group differences assessed using ANOVA tests, adjusting for sequencing depth. Beta diversity (Bray-Curtis dissimilarity matrices) and differences between groups were evaluated using PERMANOVA implemented in the *vegan:adonis2* function with 999 permutations. Differential abundance of vOTUs was evaluated using the R package MaAsLin2[81] using a linear model with total sum scaling normalization, and adjustment for age group (50–60, 60–70, and 70–80), sex, and geographic region (Bærum and Moss regions, the two recruitment regions in South-East Norway). To examine associations with diet, lifestyle, and demographic variables measured on a continuous scale, variables were grouped into tertiles. Comparisons were then made of virome variables between the lowest and highest tertiles. Participants with missing data or selecting the answer option "Unknown" (applicable to the items concerning antibiotic and antacid usage), were excluded from statistical analyses evaluating associations with diversity, composition, and differential abundance. The magnitudes of observed associations with alpha and beta diversity were quantified using Omega-squared statistics[82], which for beta diversity was calculated employing the *adonis_OmegaSq* function from the R package *micEco*. Participants with CRC diagnoses were excluded in a sensitivity analysis of associations between the gut virome and participant characteristics. Here, original effect sizes and statistical significance levels were compared with those obtained when excluding CRC cases. Custom R scripts were used for statistics and visualization of results (https://github.com/Rounge-lab/CRCbiome_virome_2023).

**Reporting summary**

Further information on research design is available in the Nature Portfolio Reporting Summary linked to this article.

## Data availability

DNA sequencing data generated in this study have been deposited in the database Federated EGA under accession code EGAS50000000170 (https://ega-archive.org/studies/EGAS50000000170). Per participant consent, submitted FASTQ files exclude reads mapping to the human genome. The data are available under restricted access due to the sensitive nature of data derived from human subjects. Processing of data from this study must comply with the General Data Protection Regulation (GDPR). Access can be obtained by following the procedure described here: https://www.mn.uio.no/sbi/english/groups/rounge-group/crcbiome/. Requests for data access can also be directed to Trine B Rounge, trinro@uio.no. The processed FASTA files of CRCbiome vOTUs detected in 5 or more individuals are available at the European Nucleotide Archive with accession number ERS16322857. The data on genome length, prevalence, taxonomy assignment of all vOTUs generated in this study, and data on their correlation to lifestyle and demographic factors, are provided in the Supplementary Data 1. A minimum dataset for reproduction of analyses is available at https://github.com/Rounge_lab/CRCbiome_virome_2023. Links to publicly available databases used in this study are provided in Supplementary Table 7.

## Code availability

The custom Snakemake pipeline and R scripts used in this study are available at: https://github.com/Rounge-lab/CRCbiome_virome_2023 (https://zenodo.org/records/10556196).

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

## Acknowledgements
The sequencing was performed at the Sequencing laboratory of Institute for Molecular Medicine Finland (FIMM) Technology Centre, University of Helsinki. We would specifically acknowledge Harri Kangas and Pekka Ellonen for their invaluable advice and help. Jan Inge Nordby and Cecilie Bucher-Johannessen contributed to sample preparation and DNA extraction. We would also like to thank Elina Vinberg and Maja Sigerseth Jakobsen for coordinating the CRCbiome project and Erik Natvig for contributing to data management. Geir Hoff, Thomas de Lange, Øyvind Holme, Kristin Randel, and Giske Ursin have contributed with establishing the CRCbiome study and nesting it to the Bowel Cancer Screening in Norway study. The Services for Sensitive Data (TSD) staff has contributed with timely solving computer issues. This project would not have been possible without funding from the Norwegian Cancer Society, projects 190179 (TBR) and 198048 (PB), the South Eastern Norway Regional Health Authority projects 2022067 (TBR), the Italian Institute for Genomic Medicine IIGM, Compagnia di San Paolo Torino, Italy, and European Union's Horizon 2020 Research and Innovation program under the Marie Sklodowska-Curie Action Grant agreement 801133 - Scientia Fellow (TBR & TR); and under the grant agreement 825410 - ONCOBIOME project (BP). The funding bodies played no role in the design of the study and collection, analysis, and interpretation of data and in writing the manuscript.

## Author contributions
TBR, PB, and TR contributed to the study conception and design. PB, TBR, ASK, EB, VB, BP, and ST participated in the data collection. PI, EB, ASK, and EA performed the data analysis. TBR, PI, EB, ASK, EA, and WMdV interpreted the results. TBR, PI, EB, ASK, and EA prepared the manuscript. All authors have read and approved the final manuscript.

## Competing interests
The authors declare no competing interests.
