## [Peer Review File · Nature Communications]

REVIEWER COMMENTS

Reviewer #1 (Remarks to the Author):

All published research articles on the fecal microbiome and especially the fecal virome have used either fresh stool samples or stool samples preserved in buffers that are designed for stabilization of the fecal microbiome. This new research paper tried to prove that the DNA virome could be analyzed in the stool collected in the FIT sampling tubes which have been widely used in national bowel cancer screening programs among different countries. Thus, once validated, the research method of this paper may enable future large-cohort clinical studies with the existing infrastructure of the bowel cancer screening program including easy transportation of FIT tube and recruitment of participants.

This paper could be improved by addressing the following issues:

1. The authors suggested that FIT tubes could be a suitable sampling method for virome characterization. However, this study only investigates the DNA virome in stool collected from FIT tubes but did not compare these data with those data collected from the current methodologies ie. either fresh stool samples or stool samples preserved in buffers that are designed for the stabilization of the fecal microbiome. Including this comparison will strengthen this hypothesis.
2. In Figure 1c, are those correlations statistically significant?
3. The supplementary data 1, 2, 3 should be labeled as supplementary table 1, 2, 3 instead as supplementary data 1, 2, 3 could not be found in the paper. Perhaps, this is just typo.
4. In Figure 1a, please add the unit of the x-axis for each panel.
5. Have any negative controls included in the study? If so, what is the process of removing the contaminants? This is important because we have to make sure that the viral signatures identified in this study do not come from the contaminations.

6. On page 9, line 271, Peduoviridae did not have a significant difference in "organic nitrogen" but the author claims that it is largely increased. This is not an appropriate description.

7. in Figure 4c, Do the genome integrations of unintegrated families have a significant decrease statistically as compared to those of the other families?

8. In Figure 5 legend, gray should be grey.

9. This paper only analyzed the fecal DNA virome. So the title should be changed from "gut virome" to "fecal DNA virome".

10. It would be good to include the Ethics approval information such as ethics approval number..etc.

11. What does the color represent in Suppl. Fig 3?

12. In Suppl Fig5, does the "original effect size" mean "Effect size with CRC"? Suppl Fig 5 also needs a longer description of what the sub-figure represents.

Reviewer #2 (Remarks to the Author):

The authors describe characterization of the virome in stool samples collected for colorectal cancer screenings. The study includes a large number of samples (1034), which is its most distinct accomplishment.

Major points:

The authors extracted DNA from samples stored in the collection buffer and demonstrate that they can then sequence the DNA and identify viral sequences. However, they authors only evaluate DNA, which should be noted in the title and abstract, since there are important RNA viruses in the gut.

The authors also include that extracting DNA from the collection buffer for sequencing works well, but there is no comparison with samples collected in a more optimal way. It is possible that the collection method has introduced biases in detection, and that limitation should be mentioned.

Some of the methods are a little unclear. It is a little unusual that libraries were size selected after pooling. The insert size is also a little larger than I might have expected for NovaSeq sequencing. What was the method of size selection on the large pool of 240 libraries? Why was that size range selected?

What databases were used for alignments? Parameters? Were translated alignments included in assigning taxonomy? Were bacterial sequences screened out? Without screening, false positives can sometimes be observed. How did the authors limit false positives?

What about human viruses? It is surprising to not see more representation of human viruses in this large data set. If these are missed because of low abundance and failure to assemble, other analysis approaches may need to be employed to ensure they are being included and the data are representative of the viral communities.

While not necessary to address this point, this paper would have more impact if the bacterial viruses were related to bacterial communities/bacterial hosts.

Minor point: Line 162, "utilizes" is written twice.

Point by point response “Exploring the gut DNA virome in fecal immunochemical test stool samples reveals novel associations with lifestyle in a large population-based study” by Istvan, Birkeland et al.

We would like to thank the reviewers for their thorough and critical revision of our manuscript. In this rebuttal letter, we address each of the concerns and suggestions raised by the Reviewers and explain how we have revised the manuscript to address these points.

Both reviewers pointed out the added value of comparing stability of FIT samples with stool samples stored in buffers designed for microbiome analysis. We agree with this assessment, and we have now included new and independent sequencing data on paired stool samples from an Italian population sampled using both the FIT kit and the Norgen nucleic acid kit. In the revised version of the manuscript, we demonstrate that virome analyses based on FIT samples provided results that are very similar to those of standard sampling procedures for gut microbiome analyses at sequencing coverages employed for in the current study (results: 124–137, figure 2c-d, supplementary figure 2). To obtain samples and conduct metagenome sequencing of the newly analyzed samples, we have relied on our collaborators Barbara Pardini and Sonia Tarallo at the Italian Institute for Genomic Medicine in Turin, who we now wish to include as co-authors for this paper. Their inclusion as co-authors has been approved by all other co-authors of the manuscript.

In addition to changes made in response to reviewers, some minor changes have been made to the manuscript, and the methods section has been moved to the end of the manuscript. We have also updated the data availability statement. All text changes from the initial version are in red font.

We have also changed the title of the manuscript according to suggestion from the reviewer:
“Exploring the gut DNA virome in fecal immunochemical test stool samples reveals novel associations with lifestyle in a large population-based study”

We hope that the revised manuscript and our responses provided below will address the issues and concerns raised by the reviewers.

REVIEWER COMMENTS

Reviewer #1 (Remarks to the Author):

All published research articles on the fecal microbiome and especially the fecal virome have used either fresh stool samples or stool samples preserved in buffers that are designed for stabilization of the fecal microbiome. This new research paper tried to prove that the DNA virome could be analyzed in the stool collected in the FIT sampling tubes which have been widely used in national bowel cancer screening programs among different countries. Thus, once validated, the research method of this paper may enable future large-cohort clinical studies with the existing infrastructure of the bowel cancer screening program including easy transportation of FIT tube and recruitment of participants.

This paper could be improved by addressing the following issues:

1. The authors suggested that FIT tubes could be a suitable sampling method for virome characterization. However, this study only investigates the DNA virome in stool collected from FIT tubes but did not compare these data with those data collected from the current methodologies i.e. either fresh stool samples or stool samples preserved in buffers that are designed for the stabilization of the fecal microbiome. Including this comparison will strengthen this hypothesis.

Response: We thank the reviewer for pointing out this limitation of our study. To address it, we have now included metagenomic sequencing data derived from stool samples from 7 individuals. The samples were collected using both FIT sampling kits and the widely used Norgen nucleic acid collection and transport tubes (Norgen Biotek Corp). These individuals were recruited in the frame of the regular Piedmont Region CRC screening program in the Microbiome and MiRNA in Torino Screening (MITOS) project (<https://doi.org/10.1053/j.gastro.2023.05.037>; <https://doi.org/10.1186/s12943-023-01869-w>). The Piedmont Region screening program invites all residents aged 59–69 to undergo a single sample biennial FIT test. Both FIT and Norgen samples were stored at -80 C for 3-5 years before DNA extraction. While these samples are limited in number, they show that usage of the FIT for virome is not inferior and very similar to that of Norgen sample collection. The results are summarized in lines 124-137 and are presented in figure 2c-d and in supplementary figure 2. Sections in materials and methods have also been added, describing these samples.

2. In Figure 1c, are those correlations statistically significant?

Response: Most associations presented in figure 2b are statistically significant. To indicate this, we have now enclosed the insignificant associations in parentheses and describe this in the figure legend in the revised version of the manuscript.

3. The supplementary data 1, 2, 3 should be labelled as supplementary table 1, 2, 3 instead as supplementary data 1, 2, 3 could not be found in the paper. Perhaps, this is just typo.

Response: We agree that the use of “supplementary data” is not appropriate, and we have changed the naming to “supplementary table”, with accompanying modifications to table numbering throughout the text. Supplementary tables 1-11 are now compiled into the Supplementary Tables.xlsx document.

4. In Figure 1a, please add the unit of the x-axis for each panel.

Response: We appreciate this suggestion. For optimal visualization purposes, the labels for the x-axes are given in the right corner of the graphs, with additional detailing of units where necessary.

5. Have any negative controls included in the study? If so, what is the process of removing the contaminants? This is important because we have to make sure that the viral signatures identified in this study do not come from the contaminations.

Response: We apologize for not having included this information in the main text of the manuscript. There were six negative controls for DNA extraction and an additional two for library preparation. The

DNA extraction controls resulted in the generation of a total of 32 QC sequencing reads, whereas sequencing of the library prep negative controls resulted in a total of 3 reads. This information has now been added to the main text (lines 481-484).

6. On page 9, line 271, *Peduviridae* did not have a significant difference in "organic nitrogen" but the author claims that it is largely increased. This is not an appropriate description.

*Response: We apologize for the confusion. What we meant to convey was that the "organic nitrogen" group of AMGs were almost exclusively found in genomes of the *Peduviridae* and those without family annotations. While no formal statistical test was reported in support of this claim, as the statistical testing was performed for one family group versus all others, the numbers are clear in this regard: among genomes of the *Peduviridae* family and those without family annotation, 13.5% ($n = 2323$ in total) carried an "organic nitrogen" AMG, whereas of the remaining genomes, 0.7% carried "organic nitrogen" AMGs ($n = 7$ in total). We have modified the text to make our intended meaning clearer (line 192).*

7. in Figure 4c, Do the genome integrations of unintegrated families have a significant decrease statistically as compared to those of the other families?

Response: Yes, these are indeed highly statistically significant. This information is now presented in Supplementary Table 5.

8. In Figure 5 legend, gray should be grey.

Response: done

9. This paper only analyzed the fecal DNA virome. So the title should be changed from "gut virome" to "fecal DNA virome".

Response: Indeed, we have only investigated the DNA virome and we revised the title accordingly. However, we would argue against using 'fecal DNA virome' instead of 'gut DNA virome'. Stool samples are a very widely used proxy for gut microbiome studies, and almost all our collective knowledge on the gut microbiome stems from the analysis of stool samples rather than invasive sampling from the gastrointestinal tract.

10. It would be good to include the Ethics approval information such as ethics approval number..etc.

Response: We apologize for not including this information in the first draft of the manuscript. The CRCbiome project was approved by the Norwegian Regional Committees for Medical and Health Research Ethics (Approval no.: 63148). The MITOS study from which paired stool samples are derived, was approved by the local Ethics committee (AOU Città della salute e della Scienza di Torino, Italy). This information is now added to the Ethical Approval section of the revised manuscript (lines 586-589).

11. What does the color represent in Suppl. Fig 3?

Response: The color represents the difference in prevalence (given in percent) of a given AMG category in vOTUs associated with a given lifestyle factor and that of all vOTUs. Red shading indicates higher prevalence in the differentially abundant vOTUs, and blue shading indicates lower prevalence in the differentially abundant vOTUs. This information is now added to the figure legend (now Supplementary Figure 5).

12. In Suppl Fig5, does the "original effect size" mean "Effect size with CRC"? Suppl Fig 5 also needs a longer description of what the sub-figure represents.

Response: Original effect size does indeed include CRC cases. We agree that there is insufficient information for the interpretation of this figure and have therefore added a more detailed description to the figure legend (now Supplementary Figure 6).

Reviewer #2 (Remarks to the Author):

The authors describe characterization of the virome in stool samples collected for colorectal cancer screenings. The study includes a large number of samples (1034), which is its most distinct accomplishment.

Major points:

The authors extracted DNA from samples stored in the collection buffer and demonstrate that they can then sequence the DNA and identify viral sequences. However, they authors only evaluate DNA, which should be noted in the title and abstract, since there are important RNA viruses in the gut.

Response: Thank you for raising this point. We have now specified that we only study DNA virome both in the title and the abstract.

The authors also include that extracting DNA from the collection buffer for sequencing works well, but there is no comparison with samples collected in a more optimal way. It is possible that the collection method has introduced biases in detection, and that limitation should be mentioned.

Response: We thank the reviewer for this suggestion. We have now added pairwise comparison between stool samples stored in the FIT buffer and in the Norgen buffer designed for DNA stabilization in fecal samples; please see also response to point 1 of Reviewer 1. Results from this comparison are presented in lines 124–137, in figure 2c-d, and in supplementary figure 2.

Some of the methods are a little unclear. It is a little unusual that libraries were size selected after pooling. The insert size is also a little larger than I might have expected for NovaSeq sequencing. What was the method of size selection on the large pool of 240 libraries? Why was that size range selected?

Response: The sequencing was performed at the FIMM Technology Centre (University of Helsinki, Finland) according to the standard library prep procedure and recommendations from the sequencing centre lab. AMPure XP (Beckman Coulter; line 478) was used for the pooled amplicons clean-up. We

did not have 240 libraries, rather each library comprised 240 samples – a number arrived upon to achieve a target of 3Gb sequencing data per sample.

What databases were used for alignments? Parameters? Were translated alignments included in assigning taxonomy? Were bacterial sequences screened out? Without screening, false positives can sometimes be observed. How did the authors limit false positives?

Response: Viral sequences were extracted from Virsorter2-predicted viral contigs using CheckV. Based on CheckV assessments, these were filtered to include only those of medium quality, high quality, or those that were categorized as complete. We used INPHARED for construction of a reference database for taxonomy assignment. INPHARED is a tool for the automatic download and filtering of complete and near-complete phage genomes from GenBank (NCBI). In the revised version of the manuscript, we further used a broader viral database that included viruses with hosts other than bacteria/archaea (see response below). vConTACT2 was used for inference of viral taxonomy, which is accomplished by construction of gene-sharing networks based on translated protein sequences. vConTACT2 was run with default parameters – details are given in line 534.

As a control for the false positives introduced at the data generation steps, we had negative controls from DNA isolation (n=6) and library preparation (n=2). The DNA extraction controls resulted in the generation of 32 QC sequencing reads, whereas sequencing of the library prep negative controls resulted in 3 reads (lines 481-484). In addition, we included two positive control samples using ZymoBIOMICS Microbial Community Standard with known composition, intended for ensuring the accuracy and quality control of microbiome analyses. Subjecting these standards to the analytic procedure used in the study resulted in the identification of 15 prophages, all assigned to taxa that infect the bacteria included in the community control (Supplementary Table 10). The fact that all viruses were specific to the community standard suggests no or low level false positive viral detection. This information is now added in lines 537-540.

What about human viruses? It is surprising to not see more representation of human viruses in this large data set. If these are missed because of low abundance and failure to assemble, other analysis approaches may need to be employed to ensure they are being included and the data are representative of the viral communities.

Response: Thank you for the suggestion. We have now included annotation to eukaryotic viruses in the updated version of the manuscript. We are unaware of any comprehensive human gut eukaryotic viral database. We therefore employed the Virus-Host database that covers RefSeq and GenBank deposited viruses and includes manually curated information on host retrieved from GenBank, RefSeq, UniProt, ViralZone and literature surveys (<https://www.genome.jp/virushostdb/>). Six vOTUs (0.02%; Supplementary Table 3) had the closest protein similarity to the eukaryotic viruses. One of them was classified as human papilloma virus 6 (CRCbiome_vOTU05832). We therefore also mapped sequencing reads to the PaVE: Papillomavirus Episteme database (<https://pave.niaid.nih.gov>). Read mapping confirmed that HPV6 was the only type of HPV identified, and that this was restricted to only one individual. Two vOTUs were also assigned taxonomy based on the clustering with the INPHARED bacteriophage database. One vOTU clustered with the Flyfo siphovirus Tbat1_6 using either database. This virus is a bacteriophage isolated from the feces of Pacific Flying Fox (<https://journals.asm.org/doi/10.1128/mra.00038-22>), and the host was erroneously annotated in the

Virus-Host database. The other vOTU was indirectly clustered both to Fadolivirus algeromassiliense (an amoeba virus targeting Vermamoeba veriformis) when using the Virus-Host database and to the Acinetobacter phage MD-2021a when annotating against the INPHARED database. Both references had low similarity to the vOTU on the nucleotide level. Given these uncertainties in host annotations, reference databases' scarcity, and generally low abundance of eukaryotic DNA viruses in the gut, we decided to focus on the phage virome. These results are now provided in lines 152–166.

While not necessary to address this point, this paper would have more impact if the bacterial viruses were related to bacterial communities/bacterial hosts.

Response: We agree with the reviewer that bacteria/viral interactions are a very interesting topic to investigate. However, this question is so broad and requires such a comprehensive description that we believe it is better addressed in a separate manuscript.

Minor point: Line 162, "utilizes" is written twice.

Response: The text has been changed, and now reads 'employs BBTools utilities' (line 498).

REVIEWER COMMENTS

Reviewer #1 (Remarks to the Author):

The authors have addressed most of my questions. However, the following issues need to be addressed:

1. Although 7 paired samples were included for the validation of the methodology, it remains unknown if these samples were collected at the same time point to minimize the confounding effects. Moreover, no statistical analyses were done on Fig 2c and 2d to prove the authors' claim that "Sample identity is more important than sampling methodology in determining the similarity of samples". Lastly, it will be good to validate the virome signatures identified in this study by comparing them with the virome signatures in the literature. This will further confirm that this methodology is valid.

2. In Figure 2b, it remains unclear which associations are significant and which are not significant. Please annotate the figure to indicate the significant associations.

3. Although the description of negative controls was included, there is no description of how these negative controls were used to perform decontamination steps for removing the contaminants from the sequencing data before performing the downstream virome analyses.

4. The title of the manuscript should still be more specific and accurate to use "Fecal or Stool DNA Virome" rather than "Gut DNA Virome" no matter if it is related to invasiveness or not. This is to differentiate it from the gut mucosal DNA virome.

5. Please provide the Ethics number for the MITOS study.

Reviewer #2 (Remarks to the Author):

The authors have addressed most comments. Inclusion of the additional samples to compare storage methods is very useful.

The questions that I previously asked about methods were not adequately addressed.

1. Responses were given in the reviewer response document but not clarified in the text of the manuscript.

2. Methods were not entirely clarified. For example, the authors should say whether the default databases that come with VirSorter2 were used (and specify versions).

3. The library construction/pooling information was not clarified in the text of the document, and the response and manuscript do not provide the same explanations. The flow chart (Fig 1) shows that each sample gets a library, and the methods say that libraries were pooled. The response says that samples were pooled before library construction, but then only a few libraries would be made, and the authors would not be able to associate viral communities at the sample-level with any of the characteristics like diet. The response also references amplicons, but there are no amplicons in metagenomic data. This still needs clarification.

Point by point response “Exploring the gut DNA virome in fecal immunochemical test stool samples reveals novel associations with lifestyle in a large population-based study” by Istvan, Birkeland et al.

We would like to thank the reviewers for their thorough and critical revision of our manuscript. Below, we provide point-by-point responses to the comments made by the reviewers.

REVIEWER COMMENTS

Reviewer #1 (Remarks to the Author):

The authors have addressed most of my questions. However, the following issues need to be addressed:

1. Although 7 paired samples were included for the validation of the methodology, it remains unknown if these samples were collected at the same time point to minimize the confounding effects. Moreover, no statistical analyses were done on Fig 2c and 2d to prove the authors' claim that "Sample identity is more important than sampling methodology in determining the similarity of samples". Lastly, it will be good to validate the virome signatures identified in this study by comparing them with the virome signatures in the literature. This will further confirm that this methodology is valid.

We agree that the sampling procedures for the paired samples is an important issue, and we have now described it in detail. Participants were recruited based on their FIT results, after which participants were asked to provide an additional stool sample using the Norgen sampling kit. This means that participants had FIT and stool samples collected 2-3 weeks apart, with the stool samples being collected at a later time point. This may, as the reviewer points out, lead to some confounding effects, however, we would argue that the time separating FIT and Norgen sampling should have led to lower measured correspondence, hence supporting our interpretation that FIT and Norgen sampling methods produce similar results.

Regarding our statement that sample identity is more important than sampling methodology, we have now included results from PERMANOVA tests supporting this claim (line 129-130).

In regard to the comment on method validity and question on whether our reported signatures are similar to published signatures, we consider our analysis of paired FIT/Norgen samples to demonstrate that using FIT tests as a sampling method is appropriate. Moreover, as is pointed out in the discussion section (lines 307-309), the degree to which identified viruses are represented in reference databases is in line with previous reports. While comparison is complicated by most earlier works relying on previous iterations of the taxonomic classification system, viral taxonomy does correspond well with published works. Lastly, to evaluate the representativeness of the data presented, we mapped reads from a public dataset employing similar sequencing methodology to the CRCbiome vOTUs (Thomas et al., Nat med, 2019), showing consistent, but somewhat lower prevalence of CRCbiome vOTUs in this population (Pearson $R^2 = 0.81$, $p < 0.001$). These results are presented in a new supplementary figure 3, and on lines 137-140.

2. In Figure 2b, it remains unclear which associations are significant and which are not significant. Please annotate the figure to indicate the significant associations.

As most associations in this figure were significant, and to avoid an overly cluttered figure, we opted for highlighting the insignificant associations rather than the significant ones. We have modified the figure caption clarifying this (line 146).

3. Although the description of negative controls was included, there is no description of how these

negative controls were used to perform decontamination steps for removing the contaminants from the sequencing data before performing the downstream virome analyses.

Response: The negative controls, including six for DNA extraction and two for library preparation, were included to monitor any potential contamination, and we did not observe any contamination in our sequencing data. The DNA extraction controls resulted in a total of 32 QC sequencing reads, and the library preparation negative controls yielded only 3 reads. Given the negligible number of reads in the negative controls, we did not consider additional steps for removing contaminants necessary.

4. The title of the manuscript should still be more specific and accurate to use "Fecal or Stool DNA Virome" rather than "Gut DNA Virome" no matter if it is related to invasiveness or not. This is to differentiate it from the gut mucosal DNA virome.

We appreciate the suggestion to stress that this paper explores the virome of fecal samples and we understand the need to distinguish mucosal microbiome community from the luminal one. We believe, however, that we do stress the sample type in the title as it states, 'Exploring the gut DNA virome in fecal immunochemical test stool samples...':

*PubMed search of all articles with key words 'gut virome' published in the last five years (2019-2023) resulted in a list of 60 articles. After exclusion of review papers and papers that focused not on the virome, but rather on individual viruses, the list was reduced to 35 original research articles. In 31 of them, the data were based on the fecal samples. The terms 'fecal virome/phageome/microbiome' were used in 4 papers collectively. In one of them, the terms 'fecal virome', 'gut virome' and 'enteric virome' were used interchangeably. The 'gut/enteric/intestinal virome/phageome' terms were used in 23 articles, and specifically 'gut virome' - in 19 articles describing the virome of fecal samples. For your convenience, we provide the table summarizing the PubMed search results to the end of this response letter (Table R1). Given the **recognizability** of the term 'gut virome' and the fact that the sample type is specified in the title regardless of whether we use 'gut virome' or 'fecal virome', we prefer to keep the title as is.*

5. Please provide the Ethics number for the MITOS study.

We apologize for this omission. The ethics number for the MITOS study is now specified in the manuscript.

Reviewer #2 (Remarks to the Author):

The authors have addressed most comments. Inclusion of the additional samples to compare storage methods is very useful.

The questions that I previously asked about methods were not adequately addressed.

1. Responses were given in the reviewer response document but not clarified in the text of the manuscript.

To the best of our efforts, we believe we have provided all relevant details in the manuscript as well as in the response letter. Some changes made in response to the reviewer's comments were, however, not referenced by line numbers in the response letter:

- Assessment and extraction of viral sequences pointed out in line 515 to 518.*

- *Construction of a reference database for taxonomy assignment described in lines 537-542.*
- *vConTACT2 parameters detailed in line 546.*
- *Versions of tools and databases provided in Supplementary Table 10.*

2. Methods were not entirely clarified. For example, the authors should say whether the default databases that come with VirSorter2 were used (and specify versions).

Thank you for pointing out this inadvertent omission on our part. We did indeed use the default VirSorter2 database. We have added this information to the revised manuscript (line 515) and to supplementary table 10. Here, we have also added a description of the CheckV database used.

3. The library construction/pooling information was not clarified in the text of the document, and the response and manuscript do not provide the same explanations.

The flow chart (Fig 1) shows that each sample gets a library, and the methods say that libraries were pooled. The response says that samples were pooled before library construction, but then only a few libraries would be made, and the authors would not be able to associate viral communities at the sample-level with any of the characteristics like diet. The response also references amplicons, but there are no amplicons in metagenomic data. This still needs clarification.

We apologize for the confusion with methodology description. For each sample, DNA was extracted and then processed using Nextera DNA Flex Library preparation kit (currently marketed as Illumina DNA Prep kit). Briefly, DNA from each sample was fragmented (fragmented and tagged), and all tagged fragments were then amplified using i5- and i7- indexed Illumina primers. As a result, metagenomic fragments from each sample have their own sample-specific i5/i7 index combination, which allows for later demultiplexing of sequencing reads. Following this amplification, fragmented, tagged and amplified DNA fragments were pooled together into libraries of 240 samples per library pool to ensure target sequencing depth for each sample. After pooling, each library pool was cleaned up using AMPure XT.

We have now added Supplementary Fig. 8 summarizing the wet-lab protocol.

Table R1 Summary of original research articles on gut virome published in 2019-2023 and deposited in PubMed (retrieved 13.12.2023)

PMID	Title	Authors	Journal/Book	Publication Year	DOI	Use fecal samples/term to describe the virome if yes
32179689	Honey bees harbor a diverse gut virome engaging in nested strain-level interactions with the microbiota	Bonilla-Rosso G, Steiner T, Wichmann F, Bexkens E, Engel P.	Proc Natl Acad Sci U S A	2020	10.1073/pnas.2000228117	no (honey bees; pulled hindguts)
33548686	Inter-vendor variance of enteric eukaryotic DNA viruses in specific pathogen free C57BL/6N mice	Rasmussen TS, Jakobsen RR, Castro-Mejía JL, Kot W, Thomsen AR, Vogensen FK, Nielsen DS, Hansen AK.	Res Vet Sci	2021	10.1016/j.rvsc.2021.01.022	no (intestinal content from mouse cecum)
35918425	Viral biogeography of the mammalian gut and parenchymal organs	Shkoporov AN, Stockdale SR, Lavelle A, Kondova I, Heuston C, Upadrasta A, Khokhlova EV, van der Kamp I, Ouwerling B, Draper LA, Langermans JAM, Paul Ross R, Hill C.	Nat Microbiol	2022	10.1038/s41564-022-01178-w	no (luminal content and mucosa; pigs)
30252582	Metagenomic analysis of intestinal mucosa revealed a specific eukaryotic gut virome signature in early-diagnosed inflammatory bowel disease	Ungaro F, Massimino L, Furfaro F, Rimoldi V, Peyrin-Biroulet L, D'Alessio S, Danese S.	Gut Microbes	2019	10.1080/19490976.2018.1511664	no (mucosa)
30169455	Enteric Virome and Bacterial Microbiota in Children With Ulcerative Colitis and Crohn Disease	Fernandes MA, Verstraete SG, Phan TG, Deng X, Stekol E, LaMere B, Lynch SV, Heyman MB, Delwart E.	J Pediatr Gastroenterol Nutr	2019	10.1097/MPG.0000000000002140	yes, 'fecal virome', 'enteric virome' and 'gut virome' interchangeably
34588600	Alpha-synuclein alters the faecal viromes of rats in a gut-initiated model of Parkinson's disease	Stockdale SR, Draper LA, O'Donovan SM, Barton W, O'Sullivan O, Volpicelli-Daley LA, Sullivan AM, O'Neill C, Hill C.	Commun Biol	2021	10.1038/s42003-021-02666-1	yes, 'fecal microbiome'

37528343	Common antibiotics, azithromycin and amoxicillin, affect gut metagenomics within a household	Chopyk J, Cobián Güemes AG, Ramirez-Sanchez C, Attai H, Ly M, Jones MB, Liu R, Liu C, Yang K, Tu XM, Abeles SR, Nelson K, Pride DT.	BMC Microbiol	2023	10.1186/s12866-023-02949-z	yes, 'gut metagenomics'
33779498	Expansion and persistence of antibiotic-specific resistance genes following antibiotic treatment	Kang K, Imamovic L, Misiakou MA, Bornakke Sørensen M, Heshiki Y, Ni Y, Zheng T, Li J, Ellabaan MMH, Colomer-Lluch M, Rode AA, Bytzer P, Panagiotou G, Sommer MOA.	Gut Microbes	2021	10.1080/19490976.2021.1900995	yes, 'virome'
33552999	Enteric Phageome Alterations in Patients With Type 2 Diabetes	Chen Q, Ma X, Li C, Shen Y, Zhu W, Zhang Y, Guo X, Zhou J, Liu C.	Front Cell Infect Microbiol	2021	10.3389/fcimb.2020.575084	yes, 'enteric phageome'
33876746	Primate phageomes are structured by superhost phylogeny and environment	Gogarten JF, Rühlemann M, Archie E, Tung J, Akoua-Koffi C, Bang C, Deschner T, Muyembe-Tamfun JJ, Robbins MM, Schubert G, Surbeck M, Wittig RM, Zuberbühler K, Baines JF, Franke A, Leendertz FH, Calvignac-Spencer S.	Proc Natl Acad Sci U S A	2021	10.1073/pnas.2013535118	yes, 'fecal phageome'
30867308	Fecal Viral Diversity of Captive and Wild Tasmanian Devils Characterized Using Virion-Enriched Metagenomics and Metatranscriptomics	Chong R, Shi M, Grueber CE, Holmes EC, Hogg CJ, Belov K, Barrs VR.	J Virol	2019	10.1128/JVI.00205-19	yes, 'fecal virome'; marsupial
33791236	Probing the "Dark Matter" of the Human Gut Phageome: Culture Assisted Metagenomics Enables Rapid Discovery	Fitzgerald CB, Shkoporov AN, Upadrasta A, Khokhlova EV, Ross RP, Hill C.	Front Cell Infect Microbiol	2021	10.3389/fcimb.2021.616918	yes, 'gut phageome'

	and Host-Linking for Novel Bacteriophages					
31600503	The Human Gut Virome Is Highly Diverse, Stable, and Individual Specific	Shkoporov AN, Clooney AG, Sutton TDS, Ryan FJ, Daly KM, Nolan JA, McDonnell SA, Khokhlova EV, Draper LA, Forde A, Guerin E, Velayudhan V, Ross RP, Hill C.	Cell Host Microbe	2019	10.1016/j.chom.2019.09.009	yes, 'gut virome'
33678150	Integrated gut virome and bacteriome dynamics in COVID-19 patients	Cao J, Wang C, Zhang Y, Lei G, Xu K, Zhao N, Lu J, Meng F, Yu L, Yan J, Bai C, Zhang S, Zhang N, Gong Y, Bi Y, Shi Y, Chen Z, Dai L, Wang J, Yang P.	Gut Microbes	2021	10.1080/19490976.2021.1887722	yes, 'gut virome'
33602058	The gut virome in Irritable Bowel Syndrome differs from that of controls	Coughlan S, Das A, O'Herlihy E, Shanahan F, O'Toole PW, Jeffery IB.	Gut Microbes	2021	10.1080/19490976.2021.1887719	yes, 'gut virome'
33853691	Temporal landscape of human gut RNA and DNA virome in SARS-CoV-2 infection and severity	Zuo T, Liu Q, Zhang F, Yeoh YK, Wan Y, Zhan H, Lui GCY, Chen Z, Li AYL, Cheung CP, Chen N, Lv W, Ng RWY, Tso EYK, Fung KSC, Chan V, Ling L, Joynt G, Hui DSC, Chan FKL, Chan PKS, Ng SC.	Microbiome	2021	10.1186/s40168-021-01008-x	yes, 'gut virome'
34010651	Stability of the human gut virome and effect of gluten-free diet	Garmaeva S, Gulyaeva A, Sinha T, Shkoporov AN, Clooney AG, Stockdale SR, Spreckels JE, Sutton TDS, Draper LA, Dutilh BE, Wijmenga C, Kurilshikov A, Fu J, Hill C, Zhernakova A.	Cell Rep	2021	10.1016/j.celrep.2021.109132	yes, 'gut virome'
33397897	Longitudinal dynamics of gut bacteriome, mycobiome and virome	Zhang F, Zuo T, Yeoh YK, Cheng FWT, Liu Q, Tang W, Cheung	Nat Commun	2021	10.1038/s41467-020-20240-x	yes, 'gut virome'

	after fecal microbiota transplantation in graft-versus-host disease	KCY, Yang K, Cheung CP, Mo CC, Hui M, Chan FKL, Li CK, Chan PKS, Ng SC.				
33852569	The gut virome of healthy children during the first year of life is diverse and dynamic	Taboada B, Morán P, Serrano-Vázquez A, Iša P, Rojas-Velázquez L, Pérez-Juárez H, López S, Torres J, Ximenez C, Arias CF.	PLoS One	2021	10.1371/journal.pone.0240958	yes, 'gut virome'
33959381	Characterization of the gut DNA and RNA Viromes in a Cohort of Chinese Residents and Visiting Pakistanis	Yan Q, Wang Y, Chen X, Jin H, Wang G, Guan K, Zhang Y, Zhang P, Ayaz T, Liang Y, Wang J, Cui G, Sun Y, Xiao M, Kang J, Zhang W, Zhang A, Li P, Liu X, Ullah H, Ma Y, Li S, Ma T.	Virus Evol	2021	10.1093/ve/veab022	yes, 'gut virome'
31800147	Higher frequency of vertebrate-infecting viruses in the gut of infants born to mothers with type 1 diabetes	Kim KW, Allen DW, Briese T, Couper JJ, Barry SC, Colman PG, Cotterill AM, Davis EA, Giles LC, Harrison LC, Harris M, Haynes A, Horton JL, Isaacs SR, Jain K, Lipkin WI, McGorm K, Morahan G, Morbey C, Pang ICN, Papenfuss AT, Penno MAS, Sinnott RO, Soldatos G, Thomson RL, Vuillermin P, Wentworth JM, Wilkins MR, Rawlinson WD, Craig ME; ENDIA STUDY GROUP.	Pediatr Diabetes	2020	10.1111/pedi.12952	yes, 'gut virome'
33845877	Isolation and characterisation of Φ crAss002, a crAss-like phage from the human gut that infects Bacteroides xylanisolvens	Guerin E, Shkoporov AN, Stockdale SR, Comas JC, Khokhlova EV, Clooney AG, Daly KM, Draper LA,	Microbiome	2021	10.1186/s40168-021-01036-7	yes, 'gut virome'

		Stephens N, Scholz D, Ross RP, Hill C.				
35998206	Individuality and ethnicity eclipse a short-term dietary intervention in shaping microbiomes and viromes	Li J, George Markowitz RH, Brooks AW, Mallott EK, Leigh BA, Olszewski T, Zare H, Bagheri M, Smith HM, Friese KA, Habibi I, Lawrence WM, Rost CL, Lédeczi Á, Eeds AM, Ferguson JF, Silver HJ, Bordenstein SR.	PLoS Biol	2022	10.1371/journal.pbio.3001758	yes, 'gut virome'
33606979	Massive expansion of human gut bacteriophage diversity	Camarillo-Guerrero LF, Almeida A, Rangel-Pineros G, Finn RD, Lawley TD.	Cell	2021	10.1016/j.cell.2021.01.029	yes, 'gut virome' public gut metagenomes, - database construction
34048535	Avian leukosis virus subgroup J infection alters viral composition in the chicken gut	Chen Y, Li HW, Cong F, Lian YX.	FEMS Microbiol Lett	2021	10.1093/femsle/fnab058	yes, 'gut virome'; chicken
35988037	Female reproduction and viral infection in a long-lived mammal	Negrey JD, Emery Thompson M, Dunn CD, Otali E, Wrangham RW, Mitani JC, Machanda ZP, Muller MN, Langergraber KE, Goldberg TL.	J Anim Ecol	2022	10.1111/1365-2656.13799	yes, 'gut virome'; chimpanzees
33970533	Dynamic changes occur in the DNA gut virome of female cynomolgus macaques during aging	Tan X, Chai T, Duan J, Wu J, Zhang H, Li Y, Huang Y, Hu X, Zheng P, Song J, Ji P, Jin X, Zhang H, Xie P.	Microbiologyopen	2021	10.1002/mbo3.1186	yes, 'gut virome'; macaques
33975974	Spinal Cord Injury Changes the Structure and Functional Potential of Gut Bacterial and Viral Communities	Du J, Zayed AA, Kigerl KA, Zane K, Sullivan MB, Popovich PG.	mSystems	2021	10.1128/mSystems.01356-20	yes, 'gut virome'; mice (murine)
33594055	Analysis of metagenome-assembled viral genomes from the human gut reveals diverse putative	Yutin N, Benler S, Shmakov SA, Wolf YI, Tolstoy I, Rayko M, Antipov D, Pevzner PA, Koonin EV.	Nat Commun	2021	10.1038/s41467-021-21350-w	yes, 'gut virome'; public data analysis

	CrAss-like phages with unique genomic features					
33781338	Thousands of previously unknown phages discovered in whole-community human gut metagenomes	Benler S, Yutin N, Antipov D, Rayko M, Shmakov S, Gussow AB, Pevzner P, Koonin EV.	Microbiome	2021	10.1186/s40168-021-01017-w	yes, 'gut virome'; public fecal metagenome data
31492655	CRISPR-Cas System of a Prevalent Human Gut Bacterium Reveals Hypertargeting against Phages in a Human Virome Catalog	Soto-Perez P, Bisanz JE, Berry JD, Lam KN, Bondy-Denomy J, Turnbaugh PJ.	Cell Host Microbe	2019	10.1016/j.chom.2019.08.008	yes, 'human virome'; public human metagenome data
32295774	Evaluating the Alimentary and Respiratory Tracts in Health and disease (EARTH) research programme: a protocol for prospective, longitudinal, controlled, observational studies in children with chronic disease at an Australian tertiary paediatric hospital	Coffey MJ, McKay IR, Doumit M, Chuang S, Adams S, Stelzer-Braid S, Waters SA, Kasparian NA, Thomas T, Jaffe A, Katz T, Ooi CY.	BMJ Open	2020	10.1136/bmjopen-2019-033916	yes, 'intestinal microbiome' - viral community
33919474	Novel Siphoviridae Bacteriophages Infecting Bacteroides uniformis Contain Diversity Generating Retroelement	Hedžet S, Rupnik M, Accetto T.	Microorganisms	2021	10.3390/microorganisms9050892	yes, 'intestinal virome'
37219409	The ecology of viruses in urban rodents with a focus on SARS-CoV-2	Fisher AM, Airey G, Liu Y, Gemmell M, Thomas J, Bentley EG, Whitehead MA, Paxton WA, Pollakis G, Paterson S, Viney M.	Emerg Microbes Infect	2023	10.1080/22221751.2023.2217940	yes+mucosa, mice, 'viruses/viral load'
32841606	The Gut Virome Database Reveals Age-Dependent Patterns of Virome Diversity in the Human Gut	Gregory AC, Zablocki O, Zayed AA, Howell A, Bolduc B, Sullivan MB.	Cell Host Microbe	2020	10.1016/j.chom.2020.08.003	yes+non-fecal, 'gut virome'; database construction

REVIEWERS' COMMENTS

Reviewer #1 (Remarks to the Author):

The authors have addressed all my questions and the quality of the paper has been significantly improved.